# Feature Selection for Discovering Distributional Treatment Effect Modifiers

Yoichi Chikahara[1,2]          Makoto Yamada[2]          Hisashi Kashima[2]

[1]NTT Communication Science Laboratories, Kyoto, Japan
[2]Kyoto University, Kyoto, Japan

## Abstract

Finding the features relevant to the difference in treatment effects is essential to unveil the underlying causal mechanisms. Existing methods seek such features by measuring how greatly the feature attributes affect the degree of the *conditional average treatment effect* (CATE). However, these methods may overlook important features because CATE, a measure of the average treatment effect, cannot detect differences in distribution parameters other than the mean (e.g., variance). To resolve this weakness of existing methods, we propose a feature selection framework for discovering *distributional treatment effect modifiers*. We first formulate a feature importance measure that quantifies how strongly the feature attributes influence the discrepancy between potential outcome distributions. Then we derive its computationally efficient estimator and develop a feature selection algorithm that can control the type I error rate to the desired level. Experimental results show that our framework successfully discovers important features and outperforms the existing mean-based method.

## 1 INTRODUCTION

When the effects of a treatment (e.g., drug administration) differ across individuals, elucidating why such heterogeneity exists is critical in many applications such as precision medicine [Lee et al., 2018], personalized education [Schochet et al., 2014], and targeted advertising [Taddy et al., 2016]. A popular approach to explaining treatment effect heterogeneity is to identify the features of an individual that are relevant to the degree of a treatment effect. For instance, to unveil the mechanism of COVID-19 vaccines, recent medical studies have sought the features related to the degree of vaccine-acquired immunity [Jabal et al., 2021].

To find such features, we need to measure how greatly the attributes of each feature influence the degree of a treatment effect. To this end, the existing methods use the *conditional average treatment effect* (CATE) that is conditioned on each feature, i.e., an average treatment effect across the individuals who have an identical attribute of each feature [Imai and Ratkovic, 2013, Tian et al., 2014, Zhao et al., 2022]. However, this average cannot capture distribution parameters other than the mean, such as the variance. As a result, if the attributes of a feature do not affect the average treatment effect but influence other distribution parameters, these mean-based methods will incorrectly conclude that the feature is unrelated to the treatment effect heterogeneity.

The goal of this paper is to propose a feature selection framework for discovering *distributional treatment effect modifiers*. To achieve this goal, we develop a feature importance measure that quantifies how greatly the attributes of each feature influence the discrepancy between the distributions of *potential outcomes*, i.e., the outcomes when an individual is treated and when not treated. We formulate this measure as a variance of the maximum mean discrepancy (MMD) [Gretton et al., 2012] between the conditional potential outcome distributions conditioned on each feature. We derive its computationally efficient estimator using a kernel approximation technique and establish a feature selection algorithm that can control the type I error rate (i.e., the proportion of false-positive results) to the desired level.

**Our contributions** are summarized as follows:

- We formulate an MMD-based feature importance measure for discovering distributional treatment effect modifiers (Section 3.2). We derive its computationally efficient weighted estimator using a kernel approximation technique (Section 3.3).

- We develop an algorithm that selects distributional treatment effect modifiers while controlling the type I error rate (Section 3.4). To evaluate significance, we perform multiple hypothesis tests based on the *p*-values computed with the conditional resampling scheme.

*Accepted for the 38th Conference on Uncertainty in Artificial Intelligence* (UAI 2022).

- We experimentally show that our method successfully finds the features related to treatment effect heterogeneity and outperforms the existing mean-based method.

## 2  PRELIMINARIES

### 2.1  PROBLEM SETUP

Suppose that we have a sample of $n$ individuals $\mathcal{D} = \{(a_i, \boldsymbol{x}_i, y_i)\}_{i=1}^n \overset{i.i.d.}{\sim} P(A, X, Y)$ for $i = 1, \ldots, n$. Here $A \in \{0, 1\}$ is a binary treatment ($A = 1$ if an individual is treated; otherwise, $A = 0$), $X = [X_1, \ldots, X_d]^\top$ is $d$-dimensional features (a.k.a. covariates), where each feature $X_m \in \mathcal{X}$ ($m = 1, \ldots, d$) takes either discrete or continuous values, and $Y \in \mathbb{R}$ is a continuous-valued outcome.[1] Here we assume that (1) features $X$ are measured before applying the treatment and observing outcome $Y$ (i.e., features $X$ are *pretreatment variables* and not *mediators* or *colliders* [Elwert and Winship, 2014]) and that (2) features $X$ contain all *confounders*, i.e., the variables that affect treatment $A$ and outcome $Y$. Note that these assumptions are standard in the existing work [Imai and Ratkovic, 2013, Zhao et al., 2022].

Given sample $\mathcal{D}$, we solve the problem of selecting the features in $X$ that influence the effect of treatment $A$ on outcome $Y$. In this problem, which features should be selected depends on the measurement scale of the treatment effect [Hernán and Robins, 2020, Chapter 4]. There are two measurement scales: additive scale $Y^1 - Y^0$ and multiplicative scale $Y^1/Y^0$, where $Y^0$ and $Y^1$ are random variables that are referred to as potential outcomes, each of which represents the outcome when $A = 0$ and when $A = 1$, respectively [Rubin, 1974]. In this study, we define the treatment effect for each individual on an additive scale as $Y^1 - Y^0$ because this scale is standard and widely used in numerous applications [Lee et al., 2018, Schochet et al., 2014, Taddy et al., 2016].

Unfortunately, we cannot observe treatment effect $Y^1 - Y^0$. This is because we cannot jointly observe two potential outcomes $Y^0$ and $Y^1$; we only observe either $Y^0$ or $Y^1$, which is obtained as $Y = (1 - A)Y^0 + AY^1$ ($A \in \{0, 1\}$). For this reason, existing methods use the average treatment effect across individuals, which can be estimated from the data.

### 2.2  MEAN-BASED APPROACHES

Many existing methods [Tian et al., 2014, Zhao et al., 2022] seek the features whose attributes affect the degree of the average treatment effect called CATE, which is defined for each feature's attribute, $X_m = x$ ($m = 1, \ldots, d$), as follows:

$$T_m(x) := \mathbb{E}[Y^1 - Y^0 \mid X_m = x]$$
$$= \mathbb{E}[Y^1 \mid X_m = x] - \mathbb{E}[Y^0 \mid X_m = x]. \quad (1)$$

---

[1]We assume $Y \in \mathbb{R}$ to use the kernel approximation technique [Rahimi et al., 2007], which is described in Section 3.3.

Table 1: Joint probability tables of potential outcomes in Example 1. Nonzero probabilities are shown in bold. Total expresses marginal potential outcome probabilities.

| $P(Y^0, Y^1 \mid X = 0)$ | | | | | $P(Y^0, Y^1 \mid X = 1)$ | | | | |
|---|---|---|---|---|---|---|---|---|---|
| $Y^0$ \ $Y^1$ | -1 | 0 | 1 | Total | $Y^0$ \ $Y^1$ | -1 | 0 | 1 | Total |
| -1 | 0 | 0 | 0 | 0 | -1 | 0 | 0 | 0 | 0 |
| 0 | **0.5** | 0 | **0.5** | **1.0** | 0 | 0 | **1.0** | 0 | **1.0** |
| 1 | 0 | 0 | 0 | 0 | 1 | 0 | 0 | 0 | 0 |
| Total | **0.5** | 0 | **0.5** | **1.0** | Total | 0 | **1.0** | 0 | **1.0** |

CATE $T_m(x)$ is an average treatment effect over the individuals who share an identical attribute, $X_m = x$. Note that this CATE is different from the one conditioned on all features $X$, which is an inference target of the recent causal inference methods [Chang and Dy, 2017, Hassanpour and Greiner, 2019, Hill, 2011, Künzel et al., 2019, Nie and Wager, 2021, Shalit et al., 2017, Yoon et al., 2018].

Using CATE $T_m$ ($m = 1, \ldots, d$), the features that influence the degree of the average treatment effect are defined as the following *treatment effect modifiers*:

**Definition 1** (Rothman et al. [2008]). *Feature $X_m$ is said to be a treatment effect modifier if there are at least two values of $X_m$, $x_m$ and $x_m^\star$ ($x_m \neq x_m^\star$), such that CATE $T_m$ in (1) takes different values, i.e., $T_m(x_m) \neq T_m(x_m^\star)$.*

Definition 1 states that feature $X_m$ is a treatment effect modifier if CATE $T_m(x)$ is not a constant with respect to value $X_m = x$. Roughly speaking, when we group individuals by their $X_m$'s values and compute the average treatment effect in each group of the individuals, if there are at least two groups with different averages, then feature $X_m$ is a treatment effect modifier [VanderWeele, 2009].

The existing methods seek such treatment effect modifiers by fitting a regression model that is linear in treatment $A$ with a sparse regularizer [Imai and Ratkovic, 2013, Sechidis et al., 2021, Tian et al., 2014, Zhao et al., 2022].

### 2.3  WEAKNESS OF MEAN-BASED APPROACHES

Since the above mean-based methods rely on the average treatment effect, they cannot detect the features whose attributes do not influence the average treatment effect but do affect other functionals of the joint distribution of potential outcomes, such as the covariance between potential outcomes and the treatment effect variance [Russell, 2021]. To illustrate such a feature, consider the following toy example:

**Example 1.** *Let $Y^0, Y^1 \in \{-1, 0, 1\} \subset \mathbb{R}$ be the potential outcomes and let $X \in \{0, 1\}$ be a binary feature. Suppose that joint distribution $P(Y^0, Y^1 \mid X)$ is given as Table 1. Then feature $X$'s values are irrelevant to the average treatment*

*effect and the covariance between potential outcomes but relevant to the treatment effect variance:*

$$\mathbb{E}[Y^1 - Y^0 \mid X = 0] = \mathbb{E}[Y^1 - Y^0 \mid X = 1] = 0$$

$$\text{Cov}[Y^0, Y^1 \mid X = 0] = \text{Cov}[Y^0, Y^1 \mid X = 1] = 0$$

$$\text{Var}[Y^1 - Y^0 \mid X = 0] = 1; \quad \text{Var}[Y^1 - Y^0 \mid X = 1] = 0.$$

Joint distribution $P(Y^0, Y^1 \mid X)$ presented in Table 1 shows that feature $X$ is related to a difference in treatment effects: While no individual with attribute $X = 1$ receives any treatment effect, those with $X = 0$ get positive or negative effects. However, since the CATE values do not depend on $X$, the existing mean-based methods will incorrectly conclude that feature $X$ is unrelated to the treatment effect heterogeneity. This implies that using CATE is insufficient to capture such *distributional* treatment effect heterogeneity and might lead to overlooking important features.

## 3  PROPOSED METHOD

### 3.1  DETECTING DISTRIBUTIONAL HETEROGENEITY

We propose a feature selection framework for discovering the features related to distributional treatment effect heterogeneity. To find such features, we consider the problem of determining whether the values of each feature $X_m$ ($m = 1, \ldots, d$) influence the functionals of the joint distribution of potential outcomes $P(Y^0, Y^1 \mid X_m)$, such as the average treatment effect, the treatment effect variance, and the covariance between potential outcomes. [2] This problem is challenging because we cannot infer joint distribution $P(Y^0, Y^1 \mid X_m)$, since we can never jointly observe potential outcomes $Y^0$ and $Y^1$ as described in Section 2.1.

To overcome this challenge, we propose measuring the importance of each feature $X_m$ ($m = 1, \ldots, d$) by quantifying how greatly $X_m$'s values influence the discrepancy between conditional distributions $P(Y^0 \mid X_m)$ and $P(Y^1 \mid X_m)$. This idea is motivated by the following fact: *if the discrepancy between $P(Y^0 \mid X_m)$ and $P(Y^1 \mid X_m)$ varies with $X_m$'s values, then joint distribution $P(Y^0, Y^1 \mid X_m)$ is also changeable depending on $X_m$'s values, and some functionals of the joint distribution depend on $X_m$.* This fact can be easily proved by taking its contraposition, as shown in Appendix A.

Such an idea enables us to detect feature $X$ in Example 1, whose values influence the treatment effect variance. This

---

[2] Identifying which functionals are affected by each feature's values is extremely challenging due to the impossibility of inferring the joint distribution. One possible solution is to use techniques for estimating the lower and upper bounds on these functionals [Chen et al., 2016, Russell, 2021, Shingaki and Kuroki, 2021]. Although such bounds require several additional assumptions, they have been successfully applied in several fields, including fairness-aware machine learning [Chikahara et al., 2021].

is because, in this example, the discrepancy between conditional potential outcome distributions $P(Y^0 \mid X)$ and $P(Y^1 \mid X)$ changes depending on $X$'s values.

Note, however, that our idea does not always work well. This is because there are counterexamples where feature $X_m$'s values do not affect the discrepancy between conditional distributions $P(Y^0 \mid X_m)$ and $P(Y^1 \mid X_m)$ but influence joint distribution $P(Y^0, Y^1 \mid X_m)$. We take a counterexample in Appendix B and present the empirical performances in such cases in Appendix E.1. Nevertheless, compared with the existing methods, we can detect a wider variety of features relevant to treatment effect heterogeneity, which leads to a better understanding of the underlying causal mechanisms.

### 3.2  FEATURE IMPORTANCE MEASURE

To express the importance of each feature $X_m$ ($m = 1, \ldots, d$), we measure the discrepancy between distributions $P(Y^0 \mid X_m)$ and $P(Y^1 \mid X_m)$ using the MMD [Gretton et al., 2012].

In fact, there are several MMD-based metrics for measuring the discrepancy between potential outcome distributions [Bellot and van der Schaar, 2021, Muandet et al., 2021, Park et al., 2021]. However, these metrics cannot be applied in our setting because they are not designed for the conditional distributions conditioned on a single feature; we give details of this reason in Section 5.

Consequently, we develop an MMD-based metric for conditional distributions $P(Y^0 \mid X_m)$ and $P(Y^1 \mid X_m)$. Let $k_Y \colon \mathbb{R} \times \mathbb{R} \to \mathbb{R}$ be a positive-definite kernel function. Then the squared MMD between the conditional distributions conditioned on feature value $X_m = x$ is defined as

$$D_m^2(x) := \text{MMD}^2(P(Y^0 \mid X_m = x), P(Y^1 \mid X_m = x))$$

$$= \mathbb{E}_{Y^0, Y^{0\prime} \mid X_m = X_m' = x}[k_Y(Y^0, Y^{0\prime})] + \mathbb{E}_{Y^1, Y^{1\prime} \mid X_m = X_m' = x}[k_Y(Y^1, Y^{1\prime})]$$

$$- 2 \, \mathbb{E}_{Y^0, Y^1 \mid X_m = x}[k_Y(Y^0, Y^1)], \tag{2}$$

where superscript prime $\prime$ denotes an independent copy of each random variable, and expectation $\mathbb{E}_{Y^0, Y^{0\prime} \mid X_m = X_m' = x}$ is taken with respect to $P(Y^0, Y^{0\prime} \mid X_m = X_m' = x)$; other expectations are taken in a similar manner. This metric has the following property: If $k_Y$ belongs to the class of kernel functions called *characteristic kernels* [Gretton et al., 2012], then squared MMD is $D_m^2(x) = 0$ if and only if $P(Y^0 \mid X_m = x) = P(Y^1 \mid X_m = x)$. Examples of characteristic kernels include the Gaussian kernel; we provide a brief overview on characteristic kernels in Appendix C.

Based on squared MMD $D_m^2$, we define the features related to distributional treatment effect heterogeneity as the following *distributional treatment effect modifiers*:

**Definition 2.** *Feature $X_m$ is said to be a distributional treatment effect modifier if there are at least two values of $X_m$, $x_m$ and $x_m^\star$ ($x_m \neq x_m^\star$), such that squared MMD $D_m^2$ in (2) takes different values, i.e., $D_m^2(x_m) \neq D_m^2(x_m^\star)$.*

In other words, feature $X_m$ is a distributional treatment effect modifier if the squared MMD between $P(Y^0 \mid X_m)$ and $P(Y^1 \mid X_m)$ varies depending on $X_m$'s values.

To detect such a variation, we formulate the importance of each feature $X_m$ as the variance of the squared MMD:

$$I_m := \mathrm{Var}[D_m^2(X_m)]. \tag{3}$$

## 3.3 ESTIMATOR OF FEATURE IMPORTANCE

To estimate feature importance measure $I_m$ in (3), we need to compute the expected values in (2) whose expectations can be represented as those over conditional distributions $P(Y^0 \mid X_m = x)$ and $P(Y^1 \mid X_m = x)$.

However, we cannot directly compute them because we have no access to the observations from these conditional distributions. To overcome this difficulty, we develop a weighted estimator that can be computed from the observed data.

### 3.3.1 Weighted Conditional MMD (WCMMD)

To infer squared MMD $D_m^2(x)$ in (2), we develop an estimator of the expected value over conditional distribution $P(Y^a \mid X_m = x)$ ($a \in \{0, 1\}$) using a weighting-based estimation technique called importance sampling.

To derive such an estimator, we use weight functions called inverse probability weights [Rosenbaum and Rubin, 1983]:

$$w^0(A, X) = \frac{\mathbf{I}(A = 0)}{1 - \mathrm{e}(X)}, \quad w^1(A, X) = \frac{\mathbf{I}(A = 1)}{\mathrm{e}(X)}, \tag{4}$$

where $\mathrm{e}(X) := P(A = 1 \mid X)$ is the conditional distribution called a *propensity score*, and $\mathbf{I}(A = a)$ is an indicator function that takes 1 if $A = a$; otherwise 0. In addition, we make the two standard assumptions: *positivity*, which imposes support condition $0 < \mathrm{e}(x) < 1$ for all $x$ [Rosenbaum and Rubin, 1983], and *conditional ignorability* (a.k.a. *strong ignorability*), which requires conditional independence relation $\{Y^0, Y^1\} \perp A \mid X$; this relation is satisfied if features $X$ are pretreatment variables, contain no mediator or collider, and include all confounders [Elwert and Winship, 2014].

Under these assumptions, for instance, expected value $\mathbb{E}_{Y^1 \mid X_m = x}[Y^1]$ can be reformulated as

$$
\begin{aligned}
&\mathbb{E}_{Y^1 \mid X_m = x}[Y^1] \\
&= \mathbb{E}_{X_{-m} \mid X_m = x}[\mathbb{E}_{Y^1 \mid X_{-m}, X_m = x}[Y^1]] \\
&= \mathbb{E}_{X_{-m} \mid X_m = x, A = 1}\left[\mathbb{E}_{Y \mid X_{-m}, X_m = x, A = 1}\left[\frac{P(A = 1)}{P(A = 1 \mid X)} Y\right]\right] \\
&= \mathbb{E}_{A, X_{-m}, Y \mid X_m = x}[w^1(A, X) Y],
\end{aligned}
$$

where $X_{-m} := X \backslash X_m$ denotes the features with $X_m$ removed.

To estimate squared MMD $D_m^2(x)$ in (2) in the same way, we formulate the following estimator, which we call a *weighted*

*conditional MMD* (WCMMD):

$$
\begin{aligned}
&\mathrm{WCMMD}_{X_m = x}^2 \\
&:= \mathbb{E}_{A, A', X_{-m}, X'_{-m}, Y, Y' \mid X_m = X'_m = x}[w^0(A, X) w^0(A', X') k_Y(Y, Y')] \\
&+ \mathbb{E}_{A, A', X_{-m}, X'_{-m}, Y, Y' \mid X_m = X'_m = x}[w^1(A, X) w^1(A', X') k_Y(Y, Y')] \\
&- 2\,\mathbb{E}_{A, A', X_{-m}, X'_{-m}, Y, Y' \mid X_m = X'_m = x}[w^0(A, X) w^1(A', X') k_Y(Y, Y')].
\end{aligned}
\tag{5}
$$

We can show that this WCMMD equals $D_m^2(x)$ under conditional ignorability and positivity assumptions:

**Proposition 1.** *Suppose that conditional ignorability and positivity hold. Then $D_m^2(x) = \mathrm{WCMMD}_{X_m = x}^2$.*

See Appendix D.1 for the proof. Hence, WCMMD has the same property with $D_m^2(x)$: If $k_Y$ is a characteristic kernel, $\mathrm{WCMMD}_{X_m = x}^2 = 0$ if and only if $P(Y^0 \mid x) = P(Y^1 \mid x)$.

### 3.3.2 Empirical Estimator of WCMMD

To infer squared MMD $D_m^2(x)$ with estimator (5), we estimate the conditional expected values conditioned on $X_m = x$ using sample $\mathcal{D} = \{(a_i, \boldsymbol{x}_i, y_i)\}_{i=1}^n \overset{i.i.d.}{\sim} P(A, X, Y)$.

If feature $X_m$ takes discrete values, we only have to take the averages over the individuals with $X_m = x$. Formally, by letting $\omega_i^{a,x}$ for $i = 1, \ldots, n$ and $a \in \{0, 1\}$ be

$$\omega_i^{a,x} = \frac{\mathbf{I}(x_{m,i} = x)}{\sum_{l=1}^n \mathbf{I}(x_{m,l} = x)} w^a(a_i, \boldsymbol{x}_i), \tag{6}$$

we can estimate the expected values in (5) by

$$
\begin{aligned}
\widehat{D_m^2}(x) &:= \sum_{i=1}^n \sum_{j=1}^n \left(\omega_i^{0,x} \omega_j^{0,x} + \omega_i^{1,x} \omega_j^{1,x}\right) k_Y(y_i, y_j) \\
&\quad - 2 \sum_{i=1}^n \sum_{j=1}^n \omega_i^{0,x} \omega_j^{1,x} k_Y(y_i, y_j).
\end{aligned}
\tag{7}
$$

For continuous-valued feature $X_m$, we smoothen indicator function $\mathbf{I}$ in (6) by employing the kernel smoothing technique [Nadaraya, 1964, Watson, 1964] as follows:

$$\omega_i^{a,x} = \frac{\frac{1}{h_{X_m}} k_{X_m}(x_{m,i}, x)}{\sum_{l=1}^n \frac{1}{h_{X_m}} k_{X_m}(x_{m,l}, x)} w^a(a_i, \boldsymbol{x}_i), \tag{8}$$

where the similarity between $X_m$'s values is measured by kernel function $k_{X_m}$ with bandwidth $h_{X_m}$; in our experiments, we formulate $k_{X_m}$ as the Gaussian kernel:

$$k_{X_m}(x_m, x_m^\star) = \exp\left(-\frac{\|x_m - x_m^\star\|^2}{h_{X_m}^2}\right).$$

In both cases where $\omega_i^{a,x}$ is given as (6) and (8), we can show the consistency of estimator $\widehat{D_m^2}(x)$, i.e., convergence to the true value in the limit of infinite sample size:

**Theorem 1.** *Suppose that weight $\omega_i^{a,x}$ is given as* (6) *or* (8). *Then under the assumptions presented in Appendix D.2, we have $\widehat{D}_m^2(x) \xrightarrow{p} D_m^2(x)$ as $n \to \infty$.*

See Appendix D.2 for the proof. In practice, we need to estimate $\omega_i^{a,x}$ by inferring propensity score $\mathrm{e}(X) := \mathrm{P}(A = 1 \mid X)$ with a regression model (e.g., neural network).

A drawback of estimator $\widehat{D}_m^2(x)$ in (7) is that it needs computation time $O(n^2)$ for sample size $n$, implying that estimating $D_m^2(x)$ for each $x = x_{m,1}, \ldots, x_{m,n}$ requires $O(n^3)$, which is impractical for large $n$. To resolve this issue, in what follows, we develop a computationally efficient variant of $\widehat{D}_m^2(x)$.

### 3.3.3 Computationally Efficient Empirical Estimator

To reduce the time of computing estimator $\widehat{D}_m^2(x)$ in (7), we employ a kernel approximation technique called random Fourier features (RFFs) [Rahimi et al., 2007].

With RFFs, we approximate kernel function $k_Y(y_i, y_j)$ in (7) as an inner product of two feature vectors:

$$k_Y(y_i, y_j) \approx \widetilde{k}_Y(y_i, y_j) = \langle z(y_i), z(y_j) \rangle_{\mathbb{R}^r}, \qquad (9)$$

where $z \colon \mathbb{R} \to \mathbb{R}^r$ is a mapping that outputs a vector of the $r$ features, where $r$ is a hyperparameter. These $r$ features are randomly sampled from the Fourier transform of kernel function $k_Y$. We formulate $k_Y$ as a Gaussian kernel with bandwidth $h_Y$; in this case, feature mapping $z$ is given as $z(y) = [\sqrt{2}\cos(\lambda_1 y + \zeta_1), \ldots, \sqrt{2}\cos(\lambda_r y + \zeta_r)]^\top$, where $\lambda_1, \ldots, \lambda_r$ are drawn from Gaussian distribution $\mathcal{N}(0, 2h_Y)$, and $\zeta_1, \ldots, \zeta_r$ are sampled from uniform distribution $\mathrm{Unif}(0, 2\pi)$, respectively [Rahimi et al., 2007].

Based on (9), we approximate estimator $\widehat{D}_m^2(x)$ in (7) as

$$\begin{aligned}\widetilde{D}_m^2(x) := {}& \langle \widetilde{\mu}_{Y^0|x}, \widetilde{\mu}_{Y^0|x}\rangle_{\mathbb{R}^r} + \langle \widetilde{\mu}_{Y^1|x}, \widetilde{\mu}_{Y^1|x}\rangle_{\mathbb{R}^r} \\ & - 2\langle \widetilde{\mu}_{Y^0|x}, \widetilde{\mu}_{Y^1|x}\rangle_{\mathbb{R}^r}\end{aligned} \qquad (10)$$

where $\widetilde{\mu}_{Y^0|x}$ and $\widetilde{\mu}_{Y^1|x}$ are the following weighted averages of the $r$-dimensional random feature vector:

$$\widetilde{\mu}_{Y^0|x} = \sum_{i=1}^n \omega_i^{0,x} z(y_i); \quad \widetilde{\mu}_{Y^1|x} = \sum_{i=1}^n \omega_i^{1,x} z(y_i).$$

Using (10), we estimate our feature importance measure as

$$\widetilde{I}_m = \frac{1}{n-1}\sum_{\iota=1}^n \left(\widetilde{D}_m^2(x_{m,\iota}) - \frac{1}{n}\sum_{\varsigma=1}^n \widetilde{D}_m^2(x_{m,\varsigma})\right)^2. \qquad (11)$$

Computing this estimator requires $O(rn^2)$, which is feasible by setting hyperparameter $r$ to a moderate value.

## 3.4 FEATURE SELECTION WITH CONDITIONAL RANDOMIZATION TEST (CRT)

Using estimated measures $\widetilde{I}_1, \ldots, \widetilde{I}_d$, we select distributional treatment effect modifiers. To achieve this, we perform multiple hypothesis tests where for each $m = 1, \ldots, d$, we consider the following null and alternative hypotheses:

$$\mathcal{H}_{0,\mathrm{m}} \colon I_m = 0 \quad \text{and} \quad \mathcal{H}_{1,\mathrm{m}} \colon I_m > 0. \qquad (12)$$

To decide whether to reject each null hypothesis $\mathcal{H}_{0,\mathrm{m}}$, we compute $p$-value $p_m$, i.e., the probability of obtaining test statistic $I_m$ such that $I_m \geq \widetilde{I}_m$ under null hypothesis $\mathcal{H}_{0,\mathrm{m}}$. Evaluating this $p$-value requires the distribution of test statistic $I_m$ under $\mathcal{H}_{0,\mathrm{m}}$. However, analytically deriving this distribution is extremely difficult because the asymptotic distributions of data-dependent weights $\omega_i^{0,x}$ and $\omega_i^{1,x}$ in feature importance measure $\widetilde{I}_m$ are unclear.

For this reason, we approximate the distribution of the test statistic under null hypothesis $\mathcal{H}_{0,\mathrm{m}}$, where feature $X_m$ is irrelevant to treatment effect heterogeneity. To this end, we simulate such an irrelevant feature for each $X_m$ without changing joint distribution $\mathrm{P}(X)$ so that the joint distribution of this synthetically generated dummy feature and other observed features $X_{-m} := X \backslash X_m$ is equal to the original joint distribution, $\mathrm{P}(X)$. To achieve this, following the resampling scheme called *conditional randomization test* (CRT) [Candes et al., 2018, Section F], we sample new $X_m$'s values from the conditional distribution, $\mathrm{P}(X_m \mid X_{-m})$, without looking at the values of treatment $A$ and outcome $Y$.

Our CRT proceeds as illustrated in Algorithm 1. We first estimate conditional distribution $\mathrm{P}(X_m \mid X_{-m})$ by fitting a generative model $\mathcal{L}$ to the data; in our experiments, we employ a widely-used deep generative model called the conditional variational autoencoder (CVAE) [Sohn et al., 2015]. Then, using fitted generative model $\mathcal{L}$, we prepare $B$ datasets, each of which contains different values of the synthetic dummy features drawn from $\mathcal{L}$. In particular, for each $b = 1, \ldots, B$, we repeat the two steps: sampling $n$ values of feature $X_m$ as $x_{m,i}^{(b)} \sim \mathcal{L}(X_m \mid \boldsymbol{x}_{-m,i})$ ($i = 1, \ldots, n$) and using these values to compute test statistic $\widetilde{I}_m^{(b)}$. By repeating these steps, we obtain an empirical distribution of the test statistic and compute a $p$-value as

$$\hat{p}_m = \frac{1}{B}\sum_{b=1}^B \mathbf{I}\left(\widetilde{I}_m^{(b)} \geq \widetilde{I}_m\right). \qquad (13)$$

After computing $p$-values $\hat{p}_1, \ldots, \hat{p}_d$, we perform multiple hypothesis tests. Since the chance of obtaining false positives increases with the number of hypotheses tested, we control such false positives by adjusting the $p$-values; we used Benjamini-Hochber (BH) adjustment procedure [Benjamini and Hochberg, 1995] in our experiments. We summarize our feature selection framework in Algorithm 2.

**Algorithm 1** Conditional Randomization Test (CRT)

---

**Input**: sample $\mathcal{D} = \{(a_i, \boldsymbol{x}_i, y_i)\}_{i=1}^n$, estimated statistic $\widetilde{I}_m$
**Output**: $p$-value $\hat{p}_m$
1: Fit generative model $\mathcal{L}$ to sample $\mathcal{D}$.
2: **for** $b = 1, \ldots, B$ **do**
3:    **for** $i = 1, \ldots, n$ **do**
4:       Draw $x_{m,i}^{(b)} \sim \mathcal{L}(X_m \mid \boldsymbol{x}_{-m,i})$.
5:       $\boldsymbol{x}_i^{(b)} \leftarrow x_{m,i}^{(b)} \cup \boldsymbol{x}_{-m,i}$
6:    **end for**
7:    Compute test statistic $\widetilde{I}_m^{(b)}$ using $\{(a_i, \boldsymbol{x}_i^{(b)}, y_i)\}_{i=1}^n$.
8: **end for**
9: Compute $p$-value $\hat{p}_m$ by Eq. (13).
10: **return** $\hat{p}_m$

---

**Algorithm 2** Proposed feature selection framework

---

**Input**: sample $\mathcal{D} = \{(a_i, \boldsymbol{x}_i, y_i)\}_{i=1}^n$, significance level $\alpha$
**Output**: feature index set $\hat{S} \subseteq \{1, \ldots, d\}$
1: **for** $m = 1, \ldots, d$ **do**
2:    Compute test statistic $\widetilde{I}_m$ with sample $\mathcal{D}$.
3:    Compute $p$-value as $\hat{p}_m \leftarrow \text{CRT}(\mathcal{D}, \widetilde{I}_m)$.
4: **end for**
5: Adjust $p$-values as $\hat{p}_1^*, \ldots, \hat{p}_d^*$ using a multiple testing procedure.
6: Select feature index set as $\hat{S} = \{m \colon \hat{p}_m^* \leq \alpha\}$.
7: **return** $\hat{S}$

---

One of the advantages of applying CRT is that if the fitted generative model equals the true conditional distribution (i.e., $\mathcal{L}(X_m \mid X_{-m}) = \text{P}(X_m \mid X_{-m})$ for all $m = 1, \ldots, d$), it can precisely control the type I error rate to be at most significance level $\alpha$ [Candes et al., 2018, Section F]. Although learning such generative models is difficult, we experimentally confirmed that our method successfully controlled the type I error rate to be close to $\alpha$ (Section 4.2).

As a disadvantage, performing CRT is computationally expensive: It requires computing the test statistic $B$ times for each feature. Although this computation is embarrassingly parallelizable, it needs $O(Bdrn^2)$ in total, even with our computationally efficient estimator of the test statistic. Our future work will investigate how to further reduce the computation time; for instance, the CRT's computationally efficient variants (e.g., Liu et al. [2021]) might be helpful.

# 4 EXPERIMENTS

## 4.1 SETUP

We compared the performance of our proposed framework with the following two baselines: (1) the existing mean-based method called the selective inference method for effect modification (SI-EM) [Zhao et al., 2022] and (2) a naive variant of our method (Naive), which samples the values of a synthetic dummy feature corresponding to $X_m$ ($m = 1, \ldots, d$) not from conditional distribution $\text{P}(X_m \mid X_{-m})$ but from (empirical) marginal distribution $\text{P}(X_m)$.

We ran all methods with significance level $\alpha = 0.05$. As regards our method and Naive, we set the number of RFFs to $r = 1000$, selected the values of kernel bandwidths $h_{X_1}, \ldots, h_{X_d}$ and $h_Y$ using a well-known heuristic called median heuristic [Schölkopf et al., 2002], and inferred propensity score $e(X)$ by fitting a feed-forward neural network that contains two linear layers with 50 neurons and Rectified Linear Unit (ReLU) activation functions. With our method, we performed a CRT by setting the number of resampled datasets to $B = 100$. Here we formulated generative model $\mathcal{L}(X_m \mid X_{-m})$ for $m = 1, \ldots, d$ as a CVAE whose encoders and decoders are given as the feed-forward neural networks that contain two linear layers with 128 neurons and ReLU functions. We confirmed that the number of neurons did not greatly affect the performance in Appendix E.2.

## 4.2 SYNTHETIC DATA EXPERIMENTS

**Data:** We prepared synthetic datasets as follows. We drew treatment $A$ from the Bernoulli distribution and features $X = [X_1, \ldots, X_d]^\top$ ($d = 30$) from the Gaussian distributions:

$$A \sim \text{Ber}(0.5),$$
$$X \mid A = 0 \sim \mathcal{N}(-\mu, \Sigma), \quad \text{and} \quad X \mid A = 1 \sim \mathcal{N}(\mu, \Sigma),$$

where Ber and $\mathcal{N}$ denote the Bernoulli and Gaussian distributions, respectively, $\mu = [0.2, \ldots, 0.2]^\top$ is a $d$-dimensional vector, and $\Sigma$ is a $d \times d$ covariance matrix whose $(i, j)$-th element is $\Sigma_{i,j} = \sigma^{|i-j|}$ ($\sigma = 0.2$) for each $i, j \in \{1, \ldots, d\}$. We sampled outcome $Y = (1 - A)Y^0 + AY^1$ by generating potential outcomes $Y^0$ and $Y^1$ with the following four generation processes where five features $X_1, \ldots, X_5$ are distributional treatment effect modifiers:

- **LinMean**:
  $$Y^0 \sim \mathcal{N}(-f(X_1, \ldots, X_5), 1); Y^1 \sim \mathcal{N}(f(X_1, \ldots, X_5), 1),$$

- **NonlinMean**:
  $$Y^0 \sim \mathcal{N}(-g(X_1, \ldots, X_5), 1); Y^1 \sim \mathcal{N}(g(X_1, \ldots, X_5), 1),$$

- **LinVar**:
  $$Y^0 \sim \mathcal{N}(-5, 1); Y^1 \sim \mathcal{N}(0, h(f(X_1, \ldots, X_5))^2),$$

- **NonlinVar**:
  $$Y^0 \sim \mathcal{N}(-5, 1); Y^1 \sim \mathcal{N}(0, h(g(X_1, \ldots, X_5))^2),$$

where $f$, $g$ and $h$ are the following functions:

$$f(X_1, \ldots, X_5) = 4X_1 + 2X_2 + X_3 + 2X_4 + 4X_5,$$
$$g(X_1, \ldots, X_5) = \sum_{j=1}^5 (X_j - 0.5)^3 + 3\sum_{j=1}^5 X_j - 6,$$
$$h(v) = \max(v, 1).$$

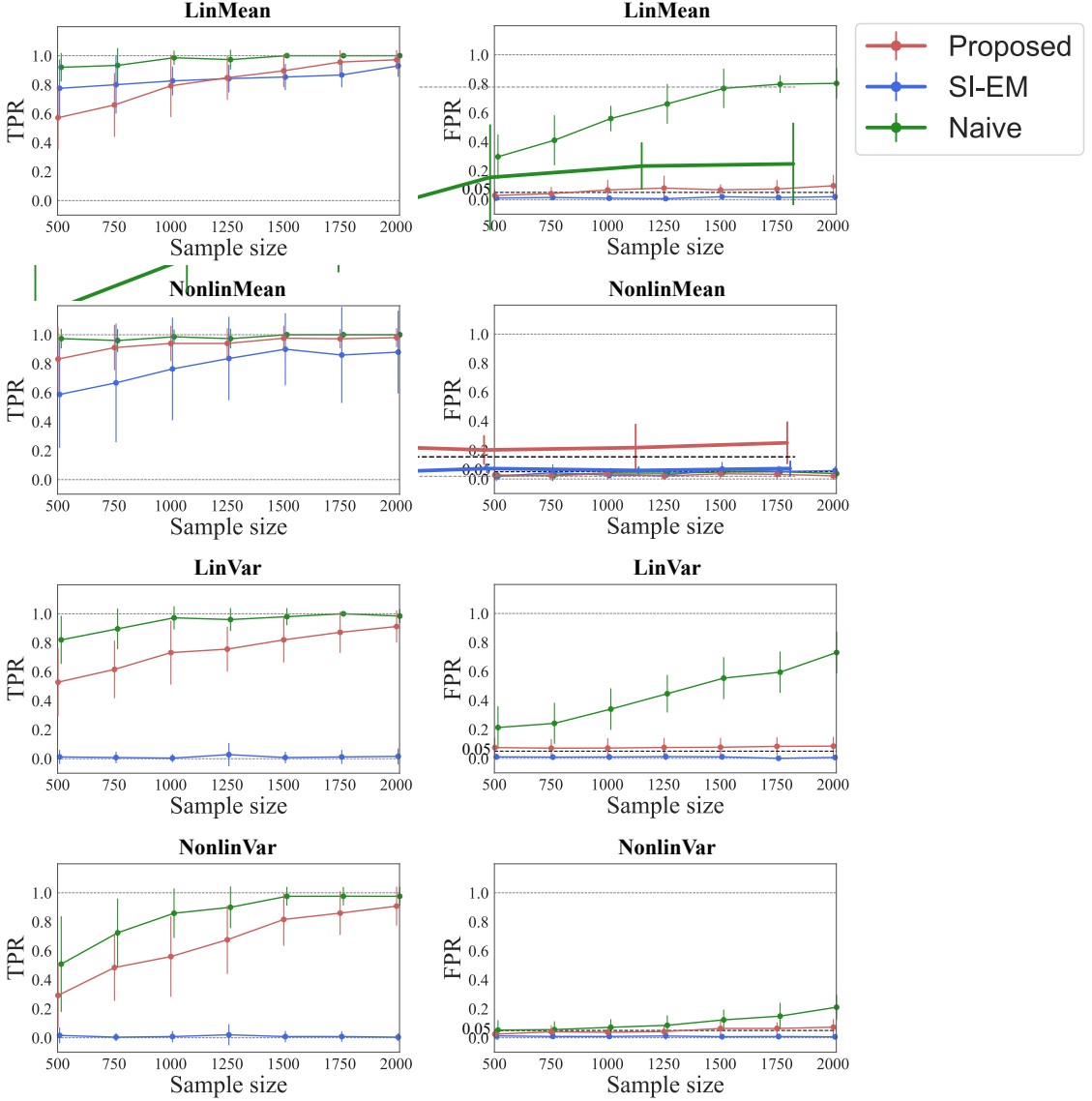

Figure 1: TPRs (left) and FPRs (right) of each method on synthetic data with sample sizes $n = 500, 750, \ldots, 2000$. Mean and standard deviation (error bars) over 50 runs with different datasets are shown.

Under LinMean and NonlinMean, features $X_1, \ldots, X_5$ influence the average treatment effect whereas under LinVar and NonlinVar, they affect the treatment effect variance.

**Results:** Using these synthetic datasets, we evaluated the performance of each method. We computed a true positive rate (TPR) and a false positive rate (FPR), defined as $\frac{d_{TP}}{d_T}$ and $\frac{d_{FP}}{d - d_T}$, where $d_T = 5$ is the number of truly relevant features, and $d_{TP}$ and $d_{FP}$ are the number of truly relevant features that are correctly selected as such and the number of irrelevant features that are wrongly selected as the relevant ones, respectively. For each method, we performed 50 experiments with different synthetic datasets generated with different random numbers and computed the average and the standard deviation of TPRs and FPRs over 50 runs.

Figure 1 presents the results on the LinMean, NonlinMean, LinVar and NonlinVar datasets. With all of them, our method successfully achieved high TPRs while controlling FPRs to be close to $\alpha = 0.05$. Although SI-EM yielded high TPRs with the LinMean and NonlinMean datasets, since this method is not designed to detect the features related to treatment effect variance, it failed to find important features from the LinVar and NonlinVar datasets. With Naive, not only the TPRs but also the FPRs were higher than our method (especially with the LinMean and LinVar datasets), indicating that it selected many features; however, many of these were false positives, which is problematic in practice.

To further illustrate the difference between our method and Naive, consider how each method approximates the $p$-value of each feature $X_m$ ($m = 1, \ldots, d$). Both methods compute

the $p$-value by sampling a synthetic dummy feature that is irrelevant to treatment effect heterogeneity; however, its sampling distribution is different. While our method samples it from (estimated) conditional distribution $P(X_m \mid X_{-m})$ in the CRT, Naive employs (empirical) marginal distribution $P(X_m)$ without looking at the values of features $X_{-m}$. The latter generation process *unnecessarily* changes joint distribution $P(X)$: The joint distribution of a synthetic feature and observed features $X_{-m}$ is greatly different from that of the original features $X$; this difference is much larger than with our method. Due to such a large change in $P(X)$, Naive failed to approximate the test statistic's distribution and yielded high FPRs. By contrast, by avoiding greatly changing joint distribution $P(X)$ with the CRT, our method effectively evaluated the statistical significance of each feature.

Meanwhile, the use of the CRT requires considerable computation time, as discussed in Section 3.4. To confirm this, we compared the run time of our method with two baselines: SI-EM and the variant of our method (Exact), which computes the feature importance measure by Eq. (7) without any approximation. Regarding our method and Exact, we evaluated the total run time, including the training time of the propensity score model and the CVAE. We ran all methods on a 64-bit CentOS machine with 2.10 GHz Xeon Gold 6130 (x2) CPUs and 256-GB RAM.

Figure 2 shows the run time on the LinMean dataset with sample sizes $n = 500, 750, \ldots, 2000$. When $n = 2000$, SI-EM and our method required 27 and 10,360 seconds, respectively, thus exhibiting a notable difference. However, our method needed far less time than Exact, demonstrating the effectiveness of kernel approximation with RFFs.

In summary, these results show the following findings:

- Our method poses a computational challenge; however, it successfully discovered the features related to the average treatment effect and the treatment effect variance.
- SI-EM does not need much time; however, it failed to find the features related to the treatment effect variance.

Thus, our proposed feature selection framework has made a significant step toward discovering the features related to distributional treatment effect heterogeneity, which, to the best of our knowledge, is the first attempt in causal inference studies. A further reduction of computation time is left as our future work, as described in Section 3.4.

## 4.3 REAL-WORLD DATA EXPERIMENTS

**Data:** We used the health records from the National Health and Nutrition Examination Survey (NHANES).[3] Following Zhao et al. [2022], we collected the records of $n = 9677$

---
[3]https://wwwn.cdc.gov/nchs/nhanes/

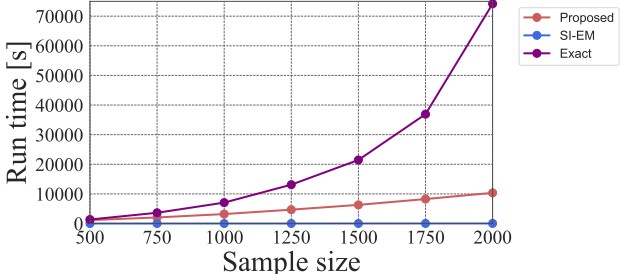

Figure 2: Run time comparison among proposed method (red), SI-EM (blue), and Exact (purple) on LinMean dataset with sample sizes $n = 500, 750, \ldots, 2000$

Table 2: $p$-values of features selected by our method from NHANES dataset: Mean and standard deviation are shown for all features with mean $p$-values less than $\alpha = 0.05$.

| Feature | Adjusted $p$-value |
|---|---|
| Age | $0.0075 \pm 0.0305$ |
| Gender | $0.0046 \pm 0.0269$ |
| Number of cigarettes smoked | $0.0 \pm 0.0$ |

individuals. Each record contains $d = 20$ features, such as age, gender, race, income, and past medical history (e.g., asthma, gout, stroke, and heart disease); 3 of them take continuous values, and the others are discrete.

With this dataset, we investigated which features modify the effects of obesity on low-grade systemic inflammation by regarding whether body mass index (BMI) exceeds 25 as treatment $A$ and serum C-reactive protein (CRP) level as outcome $Y$. Discovering such features has important medical implications because low-grade inflammation increases the risk of various chronic diseases, such as cancers and cardiovascular disease [Rodríguez-Hernández et al., 2013].

Since the truly relevant features are unknown, we cannot evaluate the TPRs and FPRs. For this reason, we compared the features selected by our method and SI-EM. Since our method is founded on the randomized algorithm (i.e., CRT), we computed the mean of the adjusted $p$-values over 50 runs and used this mean $p$-value to select the features.

**Results:** Table 2 presents the adjusted $p$-values for all features that are selected by our proposed method.

Both our method and SI-EM successfully selected age and gender, which were reported as important in the previous medical studies [Visser et al., 1999]. Although SI-EM selected only these two features, our method concluded that the number of cigarettes smoked is also statistically significant. Selecting this feature is interesting and seems reasonable because the synergistic effect of obesity and smoking on systemic inflammation has been reported in previous studies [Ólafsdóttir et al., 2005].

# 5 RELATED WORK

**Interpreting treatment effect heterogeneity:** A growing number of causal inference methods have been developed to accurately estimate heterogeneous treatment effects using neural networks [Johansson et al., 2016, Shalit et al., 2017, Yoon et al., 2018], tree-based models [Hahn et al., 2020, Hill, 2011], and machine learning frameworks called meta-learners [Künzel et al., 2019, Nie and Wager, 2021].

However, few are designed to elucidate a causal mechanism that yields the treatment effect heterogeneity. The Causal Rule Ensemble method [Lee et al., 2020] seeks the important features by learning a rule-based model that emulates the input-output relationship of a fitted treatment effect estimation model. Gilad et al. [2021] considered a hypothesis test for discovering the treatment effect modifiers from social network data. However, none of these methods can find the features related to distributional treatment effect heterogeneity because they are also based on the average treatment effect and cannot find the features related to other functionals of the joint distribution of potential outcomes.

To overcome this limitation of the existing mean-based methods, we established a feature selection framework for discovering the important features related to the functionals of the joint distribution of potential outcomes.

**MMD between potential outcome distributions:** To find distributional treatment effect modifiers, we formulated a weighted estimator of the MMD that measures the discrepancy between conditional potential outcome distributions.

Our estimator has a clear advantage in that it can consistently estimate the MMD between the conditional distributions conditioned on a single feature, $P(Y^0 \mid X_m)$ and $P(Y^1 \mid X_m)$ ($m = 1, \ldots, d$), by addressing the confounders in features $X$.

The existing estimators cannot consistently estimate such an MMD. The kernel treatment effect (KTE) [Muandet et al., 2021] and the weighted MMD (WMMD) [Bellot and van der Schaar, 2021] are designed to quantify the discrepancy between marginal distributions $P(Y^0)$ and $P(Y^1)$; hence they cannot address the conditional distributions. Although the conditional distributional treatment effect (CoDiTE) [Park et al., 2021] measures the MMD between conditional distributions $P(Y^0 \mid X)$ and $P(Y^1 \mid X)$, we cannot naively apply it by considering the setting where features $X$ only contain a single feature (i.e., $X = \{X_m\}$). This is because this measure only addresses the confounders that are included in the conditioning variables, and if setting $X = \{X_m\}$, we cannot eliminate the influence of the confounders in $X_{-m}$.

To consistently estimate the MMD between conditional distributions $P(Y^0 \mid X_m)$ and $P(Y^1 \mid X_m)$, we derived an IPW-based estimator by regarding the MMD as a function of features $X$ and then averaging out unwanted features $X_{-m}$ (by taking an integral with respect to $P(X_{-m} \mid X_m)$).

# 6 CONCLUSION

We proposed a feature selection framework for discovering the features related to the distributional treatment effect heterogeneity. The key advantage of our framework is that it can identify the features whose values influence the functionals of the joint distribution of potential outcomes if the feature values also affect the discrepancy between conditional potential outcome distributions. To the best of our knowledge, this is the first feature selection approach to revealing the causal mechanism that yields the distributional treatment effect heterogeneity. We experimentally show that our feature selection framework successfully selected important features and outperformed the existing method.

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
