# OpenReview forum: "Feature Selection for Discovering Distributional Treatment Effect Modifiers"
_auai.org/UAI/2022/Conference — UAI 2022 Poster_

### Official Review · Reviewer_oA5R · 2022-04-08

**Q2(1) Originality/Novelty:** 3
**Q2(2) Significance/Impact:** 3
**Q2(3) Correctness/Technical Quality:** 3
**Q2(6) Clarity Of Writing:** 4
**Q6 Overall Score:** 7
**Q8 Confidence In Your Score:** 4

**Q1 Summary And Contributions:**

This work describes a new method to discover the features that cause heterogeneous treatment effects across individuals. Instead of using the conventional mean-based method such as the conditional average treatment effect, this work proposes to use the variance as the measure. The key is to quantify how strongly the feature influences the discrepancy between potential outcome distributions.

**Q2 Assessment Of The Paper:**

More detailed information regarding each of these aspects is given below:

**Q2(4) Quality Of Experiments (Optional):**

2: Fair: The experimental evaluation is weak: important baselines are missing, or the results do not adequately support the main claims.

**Q2(5) Reproducibility:**

4: Excellent: Key resources (e.g., proofs, code, data) are available and key details (e.g., proof sketches, experimental setup) are comprehensively described for competent researchers to confidently and easily reproduce the main results.

**Q3 Main Strengths:**

1. A new research perspective to tackle a conventional task.
2. Well written and easy to follow.
3. Both theoretically and empirically sound.

**Q4 Main Weakness:**

1. Some experimental results do not show the effectiveness of the proposed method.
2. Discussion of the different roles of X can be helpful.

**Q5 Detailed Comments To The Authors:**

This paper addresses an important limitation of conventional approaches for finding treatment effect modifiers and introduces a novel perspective that can be potentially helpful for the research field.

Authors should consider the following to further improve the current version:
1. The only major concern I have is the results in Figure 1. The proposed method seems not to work well for Nonlin Mean and NonlinVar, i.e., it has a much lower TPR but a very similar FPR to the Naive method. This contradicts what the authors described in the text. A detailed explanation is needed.
2. It would be helpful to understand the problem and proposed method if the roles of X are discussed, e.g., confounder, mediator, or collider, in a causal graph. Would the method work in these different roles?
3. Would a method that considers both mean and variance outperform both mean-based and variance-based methods?
4. Some annotations are not described, e.g., what is B?

**Q7 Justification For Your Score:**

Overall I like the idea of this paper and it is well written. I will give an Accept if the authors can further explain the results in Figure 1.

**Q9 Complying With Reviewing Instructions:**

1: Yes.

---

### Official Review · Reviewer_cV5e · 2022-04-12

**Q2(1) Originality/Novelty:** 2
**Q2(2) Significance/Impact:** 2
**Q2(3) Correctness/Technical Quality:** 3
**Q2(6) Clarity Of Writing:** 3
**Q6 Overall Score:** 5
**Q8 Confidence In Your Score:** 4

**Q1 Summary And Contributions:**

This paper proposes a feature selection framework for discovering the features related to distributional treatment effect heterogeneity. It proposes a feature importance measure based on the discrepancy between conditional potential outcome distributions and uses the multiple  hypothesis testing method to select the important features. It also provides a detailed algorithm for the proposed method.

**Q10 Ethical Concerns (Optional):**

There is no ethical concern.

**Q2 Assessment Of The Paper:**

More detailed information regarding each of these aspects is given below:

**Q2(4) Quality Of Experiments (Optional):**

2: Fair: The experimental evaluation is weak: important baselines are missing, or the results do not adequately support the main claims.

**Q2(5) Reproducibility:**

2: Fair: Key resources (e.g., proofs, code, data) are unavailable but key details (e.g., proof sketches, experimental setup) are sufficiently well-described for an expert to confidently reproduce the main results.

**Q3 Main Strengths:**

1. The idea of the proposed method is creative.

2. The introduction of the proposed method is detailed and clear.

**Q4 Main Weakness:**

1. The computation of the proposed method is complex and heavy.

2. The true positive rate of the proposed method in the experiments is not so satisfying when the sample size is not too large.

3. The proposed method can not deal with the situations where the feature does not affect the discrepancy between conditional distributions  of potential outcomes but influence the joint distribution of potential outcomes.

**Q5 Detailed Comments To The Authors:**

1. In the problem setup, Y is assumed to be a continuous-valued outcome. But in Example 1, Y is discrete. Why Y needs to be a continuous outcome in the proposed method?

2. Based on the comment in the last paragraph of Section 3.1, there are some situations that X_m does not affect the discrepancy between conditional distributions but influence the joint distribution. Then in some situations the false negative rate will be high. Can you show the results about the false negative rate in the experiments?

3. Can you explain more about the meaning and properties of characteristic kernels in Section 3.2?

4. In the experiments, it says that inferred propensity score and the encoders and decoders of CVAE are fitted using the neural networks. How to choose the number of neurons? Whether the results are sensitive to the number of neurons?

5. Based on Figure 1, the TPRs based on  the naive methods are much higher than that based on the proposed method. Can you explain the reason for this?

6. Based on Figure 2, when the sample size is not too large (but still no smaller than 500), the TPR of the proposed method is not so satisfying. Note that in practice, the sample size of the real data can be much smaller than 500.


**Q7 Justification For Your Score:**

I make this score based on the main strengths,  weaknesses and my understanding of this paper. I think the strengths and weaknesses of this paper are comparable.

**Q9 Complying With Reviewing Instructions:**

1: Yes.

---

### Official Review · Reviewer_yGcS · 2022-04-13

**Q2(1) Originality/Novelty:** 3
**Q2(2) Significance/Impact:** 2
**Q2(3) Correctness/Technical Quality:** 3
**Q2(6) Clarity Of Writing:** 4
**Q6 Overall Score:** 7
**Q8 Confidence In Your Score:** 4

**Q1 Summary And Contributions:**

The paper presents a method for selecting covariates that may influence the variance of treatment effects. The idea is novel and the authors show in simulated experiments that the method can indeed detect such covariates in simulated data.

**Q2 Assessment Of The Paper:**

More detailed information regarding each of these aspects is given below:

**Q2(4) Quality Of Experiments (Optional):**

3: Good: The experimental evaluation is adequate, and the results convincingly support the main claims.

**Q2(5) Reproducibility:**

4: Excellent: Key resources (e.g., proofs, code, data) are available and key details (e.g., proof sketches, experimental setup) are comprehensively described for competent researchers to confidently and easily reproduce the main results.

**Q3 Main Strengths:**

The idea is novel and interesting.
The method is sound as far as I could check.
The experiments (albeit a bit limited) support the claims of the paper.

**Q4 Main Weakness:**

I do not think it is a major weakness, but it is a bit limiting that the method looks at the effect of single covariates.

**Q5 Detailed Comments To The Authors:**

Section 3.2. What is the challenge in using MMD measures designed for measuring discrepancy between potential outcomes distributions, since in Eq 2 you have conditioned on a single value of X?

Can you comment a bit on whether distributional effect modifiers would be detected using for example tests of independence to the outcome?

I would like some discussion or an example on how the method could be used to improve outcome prediction after these covariates are identified.

**Q7 Justification For Your Score:**

I believe the paper presents a novel idea and method and it is well written and justified by the results.

**Q9 Complying With Reviewing Instructions:**

1: Yes.

---

### Official Review · Reviewer_czsG · 2022-04-13

**Q2(1) Originality/Novelty:** 3
**Q2(2) Significance/Impact:** 2
**Q2(3) Correctness/Technical Quality:** 3
**Q2(6) Clarity Of Writing:** 3
**Q6 Overall Score:** 6
**Q8 Confidence In Your Score:** 4

**Q1 Summary And Contributions:**

This paper presents a feature selection framework and an algorithm to identify "distributional" treatment effect modifiers, by examining how a feature influence the discrepancy between potential outcome distributions. Most existing effect modifier detection methods are based on conditional average treatment effect (CATE), which only enables them to find features affecting the mean of treatment effect distribution, but not other distributional parameters (e.g. variance).

**Q2 Assessment Of The Paper:**

More detailed information regarding each of these aspects is given below:

**Q2(4) Quality Of Experiments (Optional):**

2: Fair: The experimental evaluation is weak: important baselines are missing, or the results do not adequately support the main claims.

**Q2(5) Reproducibility:**

3: Good: Key resources (e.g., proofs, code, data) are available and key details (e.g., proofs, experimental setup) are sufficiently well-described for competent researchers to confidently reproduce the main results.

**Q3 Main Strengths:**

1. The problem proposed and studied by the paper is interesting and well motivated.
2. The proposed feature selection framework sounds reasonable, and the authors show that the empirical estimator of WCMMD can be converged to the true value in the limit of infinite samples.
3. Attention is paid to improve the efficiency of the proposed estimator.

**Q4 Main Weakness:**

1. Related work should be enhanced.
2. More based line methods should be include to strengthen the experimental evaluation.



**Q5 Detailed Comments To The Authors:**

*Related work
The existing MMD-based metrics mentioned in the 2nd paragraph of Section 3.2 should be discussed in more detail, perhaps in the Related work section. More importantly, it is not clear why those metrics cannot be applied in the setting of this paper. Although it has been stated that these metrics are not designed for the conditional distributions conditioned on a single variable, would it be possible to adapt them for the setting "conditioned on a single variable", and is the difference between those metrics and the proposed one in this paper non-trivial, and why?

*Experiments
The paper is focused on distributional treatment effect modifiers, but the "mean" is an important functional of a treatment effect distribution, and features affecting the mean should be a target to be identified by the proposed method. Given this fact, it is necessary to compare the proposed method with existing mean-based methods to fully demonstrate the capability and performance of the proposed method.

Additionally, does the proposed method assume all covariates are pretreatment variables? How would confounders play a role/affect the functioning of the proposed method?

**Q7 Justification For Your Score:**

The problem studied by the paper is well motivated, and the proposed solution looks sound. As mentioned above, however, more details on the related work should be provided to clarify the originality/significance of the contribution. The evaluation of the method should be improved by including more baseline methods.

**Q9 Complying With Reviewing Instructions:**

1: Yes.

---

### Decision · Program_Chairs · 2022-05-15

**Decision:**

Accept (Poster)

**Comment:**

Meta Review: This work proposes a method for selecting features that are important for explaining treatment effect heterogeneity. Unlike most existing works, the proposed approach is able to detect features that drive differences in the conditional distributions of counterfactual outcomes across treatments, rather than only predicting the conditional average treatment effect.

All reviewers were positive on this work (accept, accept, weak accept, borderline accept). They praised the novelty of the proposed approach, the clarity of the writing, and the (albeit limited) experiments.

During the rebuttal period, the authors replied to the main comments of all reviewers. Two of the reviewers engaged in discussion with the authors, and both of these reviewers stated that they were satisfied with the authors' replies. The other two reviewers did not engage in the discussion, but in my view the authors adequately addressed their main concerns.